# Depression, anxiety and their comorbidity in the Swedish general population: point prevalence and the effect on health-related quality of life

Robert Johansson[1], Per Carlbring[2], Åsa Heedman[1], Björn Paxling[1] and Gerhard Andersson[1,3,4]

[1] Department of Behavioural Sciences and Learning, Linköping University, Linköping, Sweden
[2] Department of Psychology, Stockholm University, Stockholm, Sweden
[3] Department of Clinical Neuroscience, Psychiatry Section, Karolinska Institutet, Stockholm, Sweden
[4] Swedish Institute for Disability Research, Linköping University, Linköping, Sweden

Corresponding author
Robert Johansson,
robert.johansson@liu.se

## ABSTRACT

**Background.** Depression and anxiety disorders are major world-wide problems. There are no or few epidemiological studies investigating the prevalence of depression, generalized anxiety disorder and anxiety disorders in general in the Swedish population.

**Methods.** Data were obtained by means of a postal survey administered to 3001 randomly selected adults. After two reminders response rate was 44.3%. Measures of depression and general anxiety were the 9-item Patient Health Questionnaire Depression Scale (PHQ-9) and the 7-item Generalized Anxiety Disorder Scale (GAD-7). The PHQ-9 identified participants who had experienced clinically significant depression (PHQ-9 $\geq$ 10), and who had a diagnosis of major depression (defined by using a PHQ-9 scoring algorithm). Clinically significant anxiety was defined as having a GAD-7 score $\geq$ 8. To specifically measure generalized anxiety disorder, the Generalized Anxiety Disorder Questionnaire-IV (GAD-Q-IV) was used with an established cut-off. Health-related quality of life was measured using the EuroQol (EQ-5D). Experiences of treatments for psychiatric disorders were also assessed.

**Results.** Around 17.2% (95% CI: 15.1–19.4) of the participants were experiencing clinically significant depression (10.8%; 95% CI: 9.1–12.5) and clinically significant anxiety (14.7%; 95% CI: 12.7–16.6). Among participants with either clinically significant depression or anxiety, nearly 50% had comorbid disorders. The point prevalence of major depression was 5.2% (95% CI: 4.0–6.5), and 8.8% (95% CI: 7.3–10.4) had GAD. Among those with either of these disorders, 28.2% had comorbid depression and GAD. There were, generally, significant gender differences, with more women having a disorder compared to men. Among those with depression or anxiety, only between half and two thirds had any treatment experience. Comorbidity was associated with higher symptom severity and lower health-related quality of life.

**Conclusions.** Epidemiological data from the Swedish community collected in this study provide point prevalence rates of depression, anxiety disorders and their comorbidity. These conditions were shown in this study to be undertreated and

associated with lower quality of life, that need further efforts regarding preventive and treatment interventions.

## INTRODUCTION

Depression and anxiety disorders are major world-wide health problems that affect a substantial number of individuals every year (*Ebmeier, Donaghey & Steele, 2006*; *Kessler, Merikangas & Wang, 2007*). In the US National Comorbidity Survey Replication (NCS-R), lifetime prevalence of mood and anxiety disorders were 20.8% and 28.8%, respectively (*Kessler, Merikangas & Wang, 2007*). Twelve-month prevalence estimates of mood and anxiety range from 6.6% to 11.9% and 5.6% to 18.1% across surveys from Europe, Australia and the US (*Baumeister & Härter, 2007*). In Sweden in 1957, the point prevalence of depression was estimated to be 4.7%, based on data from the total population ($n = 2612$) of Lundby, a small rural area in southern Sweden (*Rorsman et al., 1990*). Using the national Swedish Twin Registry, lifetime prevalence for depression was estimated to be 13.2% among men and 25.1% among women (*Kendler et al., 2006*). In the Lundby study, lifetime prevalence for depression was 27% among men and 45% among women, when participants were followed from 1957 up to 1972 (*Rorsman et al., 1990*). Importantly, the Lundby study did not use DSM criteria for major depression, which makes comparisons to prevalence rates from other countries complicated (*Rorsman et al., 1990*). To our knowledge, there exist no up-to-date point estimates of DSM-IV depression from the Swedish general population.

For anxiety disorders, there is a 12-month prevalence study from the Swedish general population regarding panic disorder (2.2%; *Carlbring et al., 2002*) and a point prevalence study on social phobia (15.6%; *Furmark et al., 1999*). In the NCS-R, the 12-month prevalence for these disorders were 2.7% and 6.8%, respectively (*Kessler et al., 2005*). From Sweden, there exist point prevalence data on generalized anxiety disorder (GAD) from primary care (3.6%; *Allgulander & Nilsson, 2003*; *Munk-Jørgensen et al., 2006*). This can be compared to NCS-R, where 12-month prevalence of GAD was 3.1% (*Kessler et al., 2005*). However, the point prevalence of GAD in the general Swedish population seems unknown.

Comorbidity between mood and anxiety disorders is known to be common. For example, among participants in the NCS-R that had a diagnosis of major depression, 57.5% also met criteria for at least one anxiety disorder (*Kessler, Merikangas & Wang, 2007*). Other epidemiological data suggests that 59.0% of individuals with GAD fulfill criteria for major depression (*Carter et al., 2001*). This suggests that comorbidity between depression and anxiety disorders is the rule rather than the exception. Comorbidity has consistently been associated with a poorer prognosis and greater demands for professional help (*Albert et al., 2008*; *Schoevers et al., 2005*). In addition, comorbidity

between depression and anxiety seems strongly associated both with role impairment and higher symptom severity (*Kessler et al., 2003*). Generally, psychiatric comorbidity is known to affect various aspects of health-related quality of life (*Carpentier et al., 2009*; *Saarni et al., 2007*; *Sherbourne et al., 2010*).

The aim of the present study was to investigate the point prevalence of depression, GAD, anxiety disorders in general, and comorbidity in a representative Swedish sample. We also investigated the effect of depression, anxiety and comorbidity on health-related quality of life.

## MATERIALS & METHODS

This study was carried out in accordance with the STROBE initiative for reporting epidemiological data (*Von Elm et al., 2007*). Approval for the study was obtained from the Institutional Review Board at the Department of Behavioural Sciences and Learning, Psychology Section, Linköping University, Sweden. By responding to the postal survey, participants gave their consent to take part in the study.

### Sample and data collection

Data collection was conducted in the autumn of 2009. A total of 3001 participants aged 18–70 years were randomly selected from the Swedish population and address register (SPAR). This register includes all persons who are registered as residents in Sweden, both Swedish and non-Swedish citizens. Systematic sampling with a randomly chosen starting point and a fixed interval was used as sampling technique. The researchers conducting the study were not involved in the sampling.

A questionnaire (described below) was mailed by surface mail to each participant ($n = 3001$), together with a stamped return envelope. There was also an explanatory letter, in which the study was described. A web version of the survey was also constructed and all participants could choose between the paper survey and the web survey. Participants were also informed that anonymity was guaranteed and that all data collected were to be used for research purposes only. About two weeks after initially sending the letters, questionnaires and stamped return envelopes were sent out to participants who had not yet responded ($n = 2175$). Thirty-nine participants could not be reached by mail and their questionnaires were returned undelivered. In addition, two questionnaires were returned because of insufficient knowledge of Swedish. A total of 1329 (44.3%) responded and were eligible for analysis. Forty-two of these questionnaires were returned using the Internet version of the survey. In cases where a participant's response had missing values, all available data were used as long as a complete score (e.g., the total score for an instrument) could be calculated. Details of attrition can be found in Tables 1–3. Among those who responded, mean age was 46.2 years ($SD = 14.5$) and 745 participants (56.1%) were female. Fifty-two percent had post-secondary education.

### Questionnaire design and measures of mental health

The survey contained questions about demographics and treatment history. Moreover, the survey contained established measures of depression, general anxiety, GAD and

**Table 1 Demographic description of the participants.**

| | | Male | Female | Total | Test statistics |
|---|---|---|---|---|---|
| Age | Mean (SD) | 46.6 (14.3) | 45.9 (14.6) | 46.2 (14.5) | $t(1193) = 0.868, p = .39$ |
| | Min–Max | 18–68 | 18–68 | 18–68 | |
| Marital status | Married or co-habiting | 424 (72.9%) | 549 (74.2%) | 973 (73.6%) | $\chi^2(N = 1322, df = 1) = 0.30, p = .58$ |
| | Single | 158 (27.1%) | 191 (25.8%) | 349 (26.4%) | |
| Having children | Yes | 436 (74.7%) | 572 (76.8%) | 1008 (75.8%) | $\chi^2(N = 1329, df = 1) = 0.80, p = .37$ |
| Educational level | Primary | 98 (18.0%) | 99 (13.9%) | 197 (15.7%) | $\chi^2(N = 1256, df = 3) = 15.7, p < .001$ |
| | Secondary | 199 (36.6%) | 211 (29.6%) | 410 (32.6%) | |
| | Post-secondary 0–2, 9 years | 90 (16.6%) | 144 (20.2%) | 234 (18.6%) | |
| | Post-secondary 3+ years | 156 (28.7%) | 259 (36.3%) | 415 (33.0%) | |
| Experiences of treatment for worrying | Yes | 76 (13.1%) | 164 (22.3%) | 240 (18.3%) | $\chi^2(N = 1314, df = 1) = 18.1, p < .001$ |
| Experiences of any treatment for psychiatric problems | Yes | 91 (15.7%) | 197 (26.7%) | 288 (21.9%) | $\chi^2(N = 1315, df = 1) = 22.9, p < .001$ |

**Table 2 Prevalence estimates.**

| | Definition | Male | Female | Total | Test statistics |
|---|---|---|---|---|---|
| **Clinically significant depression or anxiety** | | | | | |
| Depression | PHQ-9 ≥ 10 | 8.3% (6.0–10.5) | 12.9% (10.4–15.4) | 10.8% (9.1–12.5) | $\chi^2(N = 1240, df = 1) = 6.8, p < .01$ |
| Anxiety | GAD-7 ≥ 8 | 10.7% (8.1–13.2) | 17.9% (15.1–20.7) | 14.7% (12.7–16.6) | $\chi^2(N = 1266, df = 1) = 13.0, p < .001$ |
| Comorbid depression and anxiety | PHQ-9 ≥ 10 and GAD-7 ≥ 8 | 5.7% (3.8–7.7) | 10.5% (8.1–12.8) | 8.3% (6.8–9.9) | $\chi^2(N = 1201, df = 1) = 8.8, p < .01$ |
| Any disorder | PHQ-9 ≥ 10 and/or GAD-7 ≥ 8 | 13.3% (10.4–16.1) | 20.5% (17.4–23.6) | 17.2% (15.1–19.4) | $\chi^2(N = 1201, df = 1) = 10.8, p < .001$ |
| **Major depression or Generalized anxiety disorder** | | | | | |
| Major depression | PHQ-9 algorithm | 3.4% (1.9–4.9) | 6.7% (4.8–8.6) | 5.2% (4.0–6.5) | $\chi^2(N = 1240, df = 1) = 6.82, p < .01$ |
| GAD | GAD-Q-IV ≥ 5.7 | 6.6% (4.6–8.7) | 10.6% (8.3–12.8) | 8.8% (7.3–10.4) | $\chi^2(N = 1269, df = 1) = 6.13, p < .05$ |
| Comorbid major depression and GAD | PHQ-9 algorithm and GAD-Q-IV ≥ 5.7 | 2.2% (1.0–3.5) | 3.8% (2.3–5.3) | 3.1% (2.1–4.1) | $\chi^2(N = 1195, df = 1) = 2.4, ns$ |
| Any disorder | PHQ-9 algorithm and/or GAD-Q-IV ≥ 5.7 | 8.0% (5.7–10.3) | 13.4% (10.8–16.0) | 11.0% (9.2–12.7) | $\chi^2(N = 1195, df = 1) = 8.8, p < .01$ |

**Notes.**
95% confidence intervals are given in parenthesis. PHQ-9: 9-item Patient Health Questionnaire Depression Scale; GAD-7: 7-item Generalized Anxiety Disorder Scale; GAD-Q-IV: Generalized Anxiety Disorder Questionnaire-IV.

**Table 3** EuroQol (EQ-5D) distribution (percentage) of respondents reporting no problems, moderate or severe problems in different dimensions, by gender.

| EQ-5D dimension | | Male | Female | Total | Test statistics |
|---|---|---|---|---|---|
| Mobility | No problem | 548 (94.3%) | 680 (92.6%) | 1228 (93.4%) | $\chi^2(N = 1315, df = 2) = 1.50, ns$ |
| | Moderate | 32 (5.5%) | 52 (7.1%) | 84 (6.4%) | |
| | Severe | 1 (0.2%) | 2 (0.3%) | 3 (0.2%) | |
| Self-care | No problem | 572 (98.5%) | 729 (98.8%) | 1301 (98.6%) | $\chi^2(N = 1319, df = 2) = 1.57, ns$ |
| | Moderate | 6 (1.0%) | 8 (1.1%) | 14 (1.1%) | |
| | Severe | 3 (0.5%) | 1 (0.1%) | 4 (0.3%) | |
| Usual activities | No problem | 546 (94.0%) | 657 (89.4%) | 1203 (91.4%) | $\chi^2(N = 1316, df = 2) = 11.0, p < .01$ |
| | Moderate | 28 (4.8%) | 71 (9.7%) | 99 (7.5%) | |
| | Severe | 7 (1.2%) | 7 (1.0%) | 14 (1.1%) | |
| Pain/discomfort | No problem | 372 (64.2%) | 430 (58.7%) | 802 (61.1%) | $\chi^2(N = 1312, df = 2) = 5.71, ns$ |
| | Moderate | 190 (32.8%) | 268 (36.6%) | 458 (34.9%) | |
| | Severe | 17 (2.9%) | 35 (4.8%) | 52 (4.0%) | |
| Anxiety/depression | No problem | 422 (72.8%) | 440 (60.0%) | 862 (65.7%) | $\chi^2(N = 1313, df = 2) = 23.4, p < .001$ |
| | Moderate | 143 (24.7%) | 268 (36.6%) | 411 (31.3%) | |
| | Severe | 15 (2.6%) | 25 (3.4%) | 40 (3.0%) | |

health-related quality of life (HRQoL). These measures are described below. In total, there were 28 questions in the survey. A mistake in the questionnaire design made the question regarding educational level impossible to interpret whether answers were given for ongoing or completed education.

### *Depressive symptoms*

Symptoms of depression were measured using the 9-item Patient Health Questionnaire Depression Scale (PHQ-9; *Kroenke, Spitzer & Williams, 2001*). The PHQ-9 contains 9 items, with a total score ranging from 0 to 27, where each item is scored 0 to 3 (0: Not at all; 1: Several days; 2: More than half of the days; 3: Nearly every day). Total scores of 0–4 indicate no depression, 5–9 mild depression, 10–14 moderate depression, 15–19 moderately severe depression and 20–27 severe depression (*Kroenke, Spitzer & Williams, 2001*). Psychometric properties for the PHQ-9 have been shown to be good, with an internal consistency in the range Cronbach's $\alpha = .86-.89$ and a test-retest reliability of $r = .84$ (*Kroenke et al., 2010*). As the nine items correspond to the DSM-IV criteria for depression, the PHQ-9 can be used with a diagnostic algorithm to ascertain a probable diagnosis of major depression. A diagnosis is made if five or more of the nine depressive symptoms are reported to be present (defined as having an item score of $\geq 2$), and one of the symptoms is anhedonia or depressed mood (having a score of $\geq 2$ on question 1 or 2). Another way of detecting depression is to use a cut-off score. A PHQ-9 score of $\geq 10$ is an established cut-off and seem to be the optimal balance between sensitivity (88%) and specificity (88%) when detecting depression (*Kroenke, Spitzer & Williams, 2001*). The more strict PHQ-9 score of $\geq 15$ yields a higher specificity (95%), but lower sensitivity (68%) (*Kroenke, Spitzer & Williams, 2001*).

The PHQ-9 has been shown to be a valid instrument when measuring current depression in the general population (*Kroenke et al., 2009*). It has also been used to provide prevalence estimates in population-based studies conducted in Germany (*Martin et al., 2006*), Australia (*Pirkis et al., 2009*) and Canada (*Patten & Schopflocher, 2009*). All these studies used the diagnostic algorithm mentioned above, and the study by *Pirkis et al. (2009)* did also describe prevalence rates using the PHQ-9 cut-off score 10. In this study, we used the PHQ-9 diagnostic algorithm to provide an estimate of current major depression in the general population. We also used the cut-off score of $\geq 10$ to estimate the rate of 'clinically significant depression' (*Kroenke, Spitzer & Williams, 2001*; *Pirkis et al., 2009*). The PHQ-9 was chosen as a measure of depression over the established 21-item Beck Depression Inventory-II (BDI-II; *Beck, Steer & Brown, 1996*) foremost because of length, but also due to copyright issues. Strong associations have been found between the PHQ-9 and the BDI-II in various populations, indicating a convergent validity between the instruments (*Dum et al., 2008*; *Martin et al., 2006*; *Titov et al., 2011*).

### Anxiety disorders

The 7-item Generalized Anxiety Disorder Scale (GAD-7; *Spitzer et al., 2006*) was included as a questionnaire regarding anxiety disorders in general. It is a 7-item measure, with items scored 0–3, and a total score of 21. Originally developed as a screening tool for GAD, the GAD-7 has also proved to have good sensitivity and specificity as a screener for panic disorder, social anxiety disorder and post-traumatic stress disorder (*Kroenke et al., 2007*). While the cut-off of 10 is optimal for detecting GAD, a cut-off of 8 has been found to maximize sensitivity (77%) and specificity (82%) when detecting any anxiety disorder (*Kroenke et al., 2007*). In this study, we used this cut-off ($\geq 8$) to provide an estimate of 'clinically significant anxiety'. Internal consistency for the GAD-7 is excellent (Cronbach's $\alpha = .92$) and with a good test-retest reliability of $r = .83$. In addition, convergent validity of the GAD-7 has been shown to be good, as demonstrated by its correlations to the Beck Anxiety Inventory, $r = .72$, and the anxiety dimension of SCL-90, $r = .74$ (*Kroenke et al., 2010*). Similarly to the PHQ-9, the GAD-7 has been shown to be valid in the general population (*Löwe et al., 2008*). The GAD-7 was chosen over other measures of anxiety due to its short length, free availability, but also because of its possibility to screen for anxiety disorders in general (*Löwe et al., 2008*).

### Generalized anxiety disorder

To get an estimate of the prevalence of GAD, the Generalized Anxiety Disorder Questionnaire-IV (GAD-Q-IV; *Newman, Zuellig & Kachin, 2002*) was used. It is a questionnaire that mimics the structure of the DSM-IV diagnosis of GAD. The instrument is a checklist that contains questions regarding excessive worry during the last 6 months. It also lets the participant register symptoms which are common among patients with GAD, and lets the respondent give an account over his or her worry areas or topics. The last question is about impact. A cut-off of 5.7 points (83% sensitivity and 89% specificity) on the GAD-Q-IV has been reported to indicate the presence of GAD (*Newman, Zuellig & Kachin, 2002*). *Newman, Zuellig & Kachin (2002)* proposed a cut-off of 9.0 points (70%

sensitivity and 96% specificity) to get greater certainty that only persons with GAD are identified. In this study, respondents were instructed to skip the rest of the GAD-Q-IV questions if they answered "no" to the first question about experience of excessive worry which might have lead to existing GAD cases not being identified. Skipped items were coded as zero. The cut-off score of 5.7 was used in this study to provide an estimate of the prevalence of GAD.

### Health-related quality of life

The EuroQol (EQ-5D; *EuroQol Group, 1990*) was used to assess health-related quality of life (HRQoL). The instrument contains five dimensions: mobility, self-care, usual activities, pain/discomfort and anxiety/depression where respondents rate their current health on a three-point scale (*EuroQol Group, 1990*). In addition, there exists a mean of converting the scores into a single index score, using a 'tariff' (*Dolan, 1997*). This index score ranges from $-0.594$ to 1, where full health is 1 and being dead is 0 (a negative value represent a health condition considered worse than death). As there is no tariff for Sweden, we used the tariff from the UK (*Dolan, 1997*) which is commonly used in research from Sweden (*Burström, Johannesson & Diderichsen, 2001*).

## RESULTS

Demographic data are illustrated in Table 1. There were some gender differences in this data, as women (56.5% had post-secondary education) were more educated than men (45.3% had post-secondary education). Women also had more experience of treatments for worry and for psychiatric problems in general. Moreover, the rate of experience of treatment for any psychiatric disorder was 52.2% (95% CI: 45.3–59.1; $n = 107$) among individuals with clinically significant depression or anxiety ($n = 205$), and 64.9% (95% CI: 56.7–73.1; $n = 85$) among those with major depression or GAD ($n = 131$).

### Prevalence of depression and anxiety disorders

Prevalence rates are illustrated in Table 2. The estimate of clinically significant depression (PHQ-9 $\geq$ 10) was about double (10.8%) that of major depression (5.2%), using the diagnostic algorithm. Prevalence rates of anxiety disorders in general and of GAD were 14.7% and 8.8%, respectively. There were significant gender differences for all estimates, as seen in Table 2. Average scores on the PHQ-9 and the GAD-7 in the entire sample were 3.70 ($n = 1240$; 95% CI: 3.44–3.96) and 3.59 ($n = 1266$; 95% CI: 3.36–3.81) respectively. On the PHQ-9, women scored higher, 4.21 ($n = 683$; 95% CI: 3.84–4.57), than men, 3.07 ($n = 557$; 95% CI: 2.72–3.42), $t(1238) = 4.29, p < .001$. This was similar for the GAD-7, with women scoring 4.13 ($n = 704$; 95% CI: 3.81–4.45) and men scored 2.91 ($n = 562$; 95% CI: 2.62–3.21), $t(1264) = 5.33, p < .001$.

### Prevalence rates using stricter definitions

As a mean of having larger certainty that only individuals with the disorder were identified, more strict cut-offs on the PHQ-9 ($\geq$ 15) and the GAD-Q-IV ($\geq$ 9.0) were employed. This resulted in prevalence rates of 4.4% (95% CI: 3.3–5.6) for depression/depressive symptoms and 4.7% (95% CI: 3.5–5.8) for GAD.

## Prevalence of health-related quality of life problems

The most prevalent HRQoL problems were reported in the Pain/discomfort dimension where 38.9% of the participants who reported moderate or severe problems. This was followed by the Anxiety/depression dimension where the corresponding figure was 34.3%. Prevalence of moderate/severe problems in the other dimensions were less than 10%. Gender differences were observed in the dimensions Usual activities and Anxiety/depression. Details are presented in Table 3. The mean EQ-5D index value was 0.84 (0.83–0.85) for all participants ($n = 1295$). There was a significant difference where men reported higher HRQoL (0.87; $n = 572$; 95% CI: 0.85–0.88) compared to women (0.83; $n = 723$; 95% CI: 0.81–0.84), $t(1293) = 3.46$, $p < .001$.

## Comorbidity

As seen in Table 2, the rate of comorbidity was 8.3% (95% CI: 6.8–9.9) and 3.1% (95% CI: 2.1–4.1), when using the two different definitions. Put differently, among those with clinically significant depression or anxiety, 48.3% (100/207) had comorbid depression and anxiety, while the same figure was 28.2% (37/131) among those with major depression or GAD. This is illustrated in further detail in Fig. 1.

Patients with comorbid clinically significant depression and anxiety had higher symptom severity, compared to those with a single diagnosis as measured by the PHQ-9 (14.91 compared to 8.21) and the GAD-7 (13.00 compared to 8.55). The same was true for those with comorbid major depression and GAD (PHQ-9 score 18.81, GAD-7 score 14.17), compared to those with either of the diagnoses (PHQ-9 score 10.55, GAD-7 score 9.99). All the differences were statistically significant (all $t$'s > 5.13, all $p$'s < .001).

## Effects of depression, anxiety and their comorbidity on health-related quality of life

HRQoL as measured by EQ-5D index values was consistently lower among participants with disorders than those without: Clinically significant depression, 0.54 vs 0.88; clinically significant anxiety, 0.61 vs 0.88; major depression, 0.43 vs 0.87; generalized anxiety disorder, 0.63 vs 0.86. All differences were significant (all $t$'s > 11.8, all $p$'s < .001).

Among participants with comorbid depression and anxiety, compared to those with a single disorder, there were further differences in health-related quality of life. Mean EQ-5D index values were 0.51 ($n = 99$; 95% CI: 0.45–0.58) and 0.71 ($n = 105$; 95% CI: 0.66–0.75) for comorbid/not comorbid clinically significant depression and anxiety, and 0.45 ($n = 35$; 95% CI: 0.36–0.54) and 0.63 ($n = 90$; 95% CI: 0.57–0.69) for participant with comorbid major depression and GAD compared to those with a single diagnosis. These differences were significant (both $t$'s > 2.96, both $p$'s < .01). Significant differences due to comorbidity were also observed in EQ-5D dimensions Usual activities and Anxiety/depression for both definitions of comorbidity, and also for Pain/discomfort for clinically depressed or anxious participants. Details of these analyses are presented in Table 4.

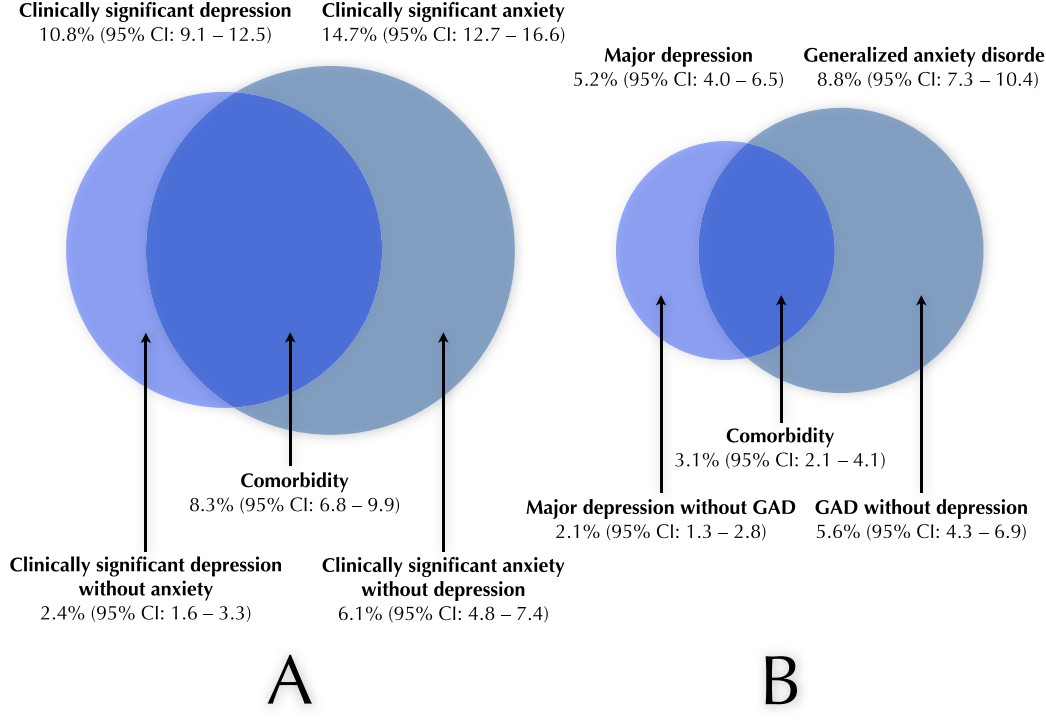

**Figure 1  Illustration of prevalence of depression, anxiety and comorbidity.** (A) Prevalence rates of clinically significant depression (134/1240), clinically significant anxiety (186/1266) and their comorbidity (100/1201). Also illustrated are clinically significant depression without anxiety (30/1245) and clinically significant anxiety without depression (77/1261). (B) Prevalence rates of current major depression (65/1240), GAD (112/1269) and their comorbidity (37/1195). Furthermore, current major depression without GAD (26/1266) and GAD without major depression (71/1267). Note: As the number of respondents were different for various measures, different prevalence rates in the figure may not add up to exactly the same figure as that given. GAD, Generalized anxiety disorder.

## DISCUSSION

This study suggest that, at any given time point, around 17.2% of the Swedish general population are experiencing clinically significant depression (10.8%) or anxiety (14.7%) that likely affect their daily lives. Around 5.2% experienced a current major depressive episode and 8.8% had generalized anxiety disorder. Half of those with either clinically significant depression or anxiety also had a comorbid disorder. Comorbidity among disorders was associated with higher symptom severity and lower health-related quality of life.

Between half and two thirds of participants with a disorder had treatment experiences. This is in line with what was found in the US National Comorbidity Survey Replication (NCS-R), where only 51.6% (95% CI: 46.1–57.2) of 12-month cases received health care treatment for depression, and where only 21.7% (95% CI: 18.1–25.2) were adequately treated (*Kessler et al., 2003*). Furthermore, there are data from a community survey in Finland, where the use of health services for mental health was investigated (*Hämäläinen et al., 2008*). Among individuals with depression, anxiety disorders, or both, only 34%, 36%, and 59% used health services, respectively. In summary, data from the present study

**Table 4** EuroQol (EQ-5D) distribution (percentage) among participant with depression and anxiety disorders, by comorbidity and single diagnosis.

| EQ-5D dimension | | Clinically significant depression or anxiety | | | | Major depression or GAD | | | |
|---|---|---|---|---|---|---|---|---|---|
| | | Comorbidity | No comorbidity | Total | Test statistics | Comorbidity | No comorbidity | Total | Test statistics |
| Mobility | No problem | 82 (82.8%) | 95 (88.8%) | 177 (85.9%) | $\chi^2(N=206, df=2) = 1.57, ns$ | 32 (88.9%) | 78 (83.9%) | 110 (85.3%) | $\chi^2(N=129, df=2) = .60, ns$ |
| | Moderate | 16 (16.2%) | 11 (10.3%) | 27 (13.1%) | | 4 (11.1%) | 13 (14.0%) | 17 (13.2%) | |
| | Severe | 1 (1.0%) | 1 (0.9%) | 2 (1.0%) | | 0 (0.0%) | 2 (2.2%) | 2 (1.6%) | |
| Self-care | No problem | 92 (92.0%) | 103 (97.2%) | 195 (94.7%) | $\chi^2(N=206, df=2) = 3.23, ns$ | 34 (91.9%) | 89 (95.7%) | 123 (94.6%) | $\chi^2(N=130, df=2) = .82, ns$ |
| | Moderate | 7 (7.0%) | 2 (1.9%) | 9 (4.4%) | | 2 (5.4%) | 3 (3.2%) | 5 (3.8%) | |
| | Severe | 1 (1.0%) | 1 (0.9%) | 2 (1.0%) | | 1 (2.7%) | 1 (1.1%) | 2 (1.5%) | |
| Usual activities | No problem | 57 (57.0%) | 87 (81.3%) | 144 (69.6%) | $\chi^2(N=207, df=2) = 15.4, p < .001$ | 15 (40.5%) | 69 (74.2%) | 8.4 (64.6%) | $\chi^2(N=130, df=2) = 15.6, p < .001$ |
| | Moderate | 37 (37.0%) | 15 (14.0%) | 52 (25.1%) | | 20 (54.1%) | 18 (19.4%) | 38 (29.2%) | |
| | Severe | 6 (6.0%) | 5 (4.7%) | 11 (5.3%) | | 2 (5.4%) | 6 (6.5%) | 8 (6.2%) | |
| Pain/ discomfort | No problem | 25 (25.0%) | 48 (45.3%) | 73 (35.4%) | $\chi^2(N=206, df=2) = 10.7, p < .01$ | 10 (27.0%) | 37 (39.8%) | 47 (36.2%) | $\chi^2(N=130, df=2) = 2.3, ns$ |
| | Moderate | 57 (57.0%) | 49 (46.2%) | 106 (51.5%) | | 22 (59.5%) | 42 (45.2%) | 64 (49.2%) | |
| | Severe | 18 (18.0%) | 9 (8.5%) | 27 (13.1%) | | 5 (13.5%) | 14 (15.1%) | 19 (14.6%) | |
| Anxiety/ depression | No problem | 2 (2.0%) | 13 (12.1%) | 15 (7.2%) | $\chi^2(N=207, df=2) = 28.8, p < .001$ | 0 (0.0%) | 5 (5.3%) | 5 (3.8%) | $\chi^2(N=130, df=2) = 19.6, p < .001$ |
| | Moderate | 66 (66.0%) | 88 (82.2%) | 154 (74.4%) | | 18 (50.0%) | 76 (80.9%) | 94 (72.3%) | |
| | Severe | 32 (32.0%) | 6 (5.6%) | 38 (18.4%) | | 18 (50.0%) | 13 (13.8%) | 31 (23.8%) | |

**Notes.**

'Clinically significant depression and anxiety' was defined as having a score of at least 10 on either or both of the PHQ-9 and the GAD-7. 'Major depression or GAD' was defined as either having a diagnosis of major depression using the PHQ-9 algorithm, or having a score of at least 5.7 on the GAD-Q-IV, or both. GAD: Generalized Anxiety Disorder. PHQ-9: 9-item Patient Health Questionnaire Depression Scale; GAD-7: 7-item Generalized Anxiety Disorder Scale; GAD-Q-IV: Generalized Anxiety Disorder Questionnaire-IV.

are in line with treatment consumption studies from other countries, and indicate that depression and anxiety disorders are undertreated conditions also in the Swedish general population.

This study estimated the point prevalence of major depression in Sweden to be 5.2%. This figure is in line with the 12 month prevalence rates from the US (6.7%, 95% CI: 6.1–7.3; *Kessler et al., 2005*, Australia (6.3%, 95% CI: 5.7–6.9; *Andrews, Henderson & Hall, 2001*) and the Netherlands (5.8%, 95% CI: 5.2–6.4; *Bijl, Ravelli & Van Zessen, 1998*). The figure 5.2% is also close to the original estimate of 4.7%, from the 1957 population in the Lundby study (*Rorsman et al., 1990*). When using a stricter definition (only 5% risk of detecting false positives), 4.4% of individuals were identified as having major depression. This and other prevalence estimates provide further indications that 5.2% is indeed a valid estimate of current point prevalence of major depression in Sweden, and therefore complements epidemiological research from the Lundby study (*Rorsman et al., 1990*) and the national Swedish Twin Registry (*Kendler et al., 2006*).

A cut-off of PHQ-9 $\geq$ 10 was used to identify clinically significant depression, and was found among 10.8% of individuals in this study. This rate is similar or somewhat higher compared to previous estimates using the same definition in the US (8.6%; *Kroenke et al., 2009*) and in Canada (8.4%; *Patten & Schopflocher, 2009*). Our result on clinically significant depression seems to mirror estimates of 'any mood disorder', e.g., from the NCS-R (*Kessler et al., 2005*) where this was found among 9.5% (95% CI: 8.7–10.3).

We found the point prevalence of clinically significant anxiety in Sweden to be 14.7%, using the definition of GAD-7 $\geq$ 8. This seems similar or somewhat higher than previous estimates from Germany (12.1%; *Löwe et al., 2008*). The mean GAD-7 score in the present study (3.59, 95% CI: 3.36–3.81) was higher than the average score from the German population study (2.95, 95% CI: 2.85–3.04) (*Löwe et al., 2008*). The rate of clinically significant anxiety found in this study can be compared with that of 18.1% for any anxiety disorder from the NCS-R (*Kessler et al., 2005*). Importantly, the GAD-7 is known only to identify GAD, social phobia, panic disorder and PTSD (*Kroenke et al., 2007*). The NCS-R estimate did also include specific phobia, which was found in 8.7% of US 12 month cases (*Kessler et al., 2005*). In conclusion, our estimates of prevalence of general anxiety seem comparable to other epidemiological studies.

Our estimate of GAD prevalence, 8.8%, using the definition of GAD-Q-IV $\geq$ 5.7 (*Newman, Zuellig & Kachin, 2002*), seem significantly higher than in other epidemiological studies from e.g., US (3.1%; *Kessler et al., 2005*) and Australia (2.6%; *Andrews, Henderson & Hall, 2001*). The stricter definition of GAD-Q-IV $\geq$ 9.0 (95% specificity) gave an estimate of 4.7%, that indicates that the true prevalence in Sweden is probably at least that high. Importantly, as noted by Kessler and colleagues (*Kessler, Walters & Wittchen, 2004*) in a review of the epidemiology of GAD, there are uncertainties about the basic epidemiological characteristics of GAD that could lead to underestimates of the true prevalence of GAD in the general population. There are known complicating factors due to changing DSM criteria over time, and the fact that DSM definitions of GAD seem to exclude a substantial

amount of individuals with chronic worry, tension and nervousness (*Kessler, Walters & Wittchen, 2004*). *Kessler, Walters & Wittchen (2004)* argued that the true current prevalence of GAD in the community could actually be as high as 5 to 8%. Our study has provided indications that the prevalence of GAD in the Swedish general population is probably within this interval.

The EQ-5D index value obtained in this study (0.84) was identical to that obtained in a study that validates the EQ-5D in the Swedish general population (*Burström, Johannesson & Diderichsen, 2001*). Absence of reported confidence intervals in *Burström, Johannesson & Diderichsen (2001)* prohibits a more detailed comparison, but when comparing HRQoL data from this study (Table 3) to Swedish normative data, no dimension differs more than 5.4%. This gives indications that HRQoL data collected in this study are overall similar to previous collected data from Sweden.

There are methodological limitations that need to be considered. In this study, we used self-reporting, rather than a diagnostic interview, such as the SCID (*First et al., 1997*) or the CIDI (*Robins et al., 1988*). While the measures used in this study are thoroughly validated (*Kroenke, Spitzer & Williams, 2001*; *Newman, Zuellig & Kachin, 2002*; *Spitzer et al., 2006*), also for use in the general population (*Kroenke et al., 2009*; *Löwe et al., 2008*) we still consider this as a limitation. A further limitation is that with the data collected on educational level it was impossible to interpret whether a respondent had a certain education ongoing or completed. This makes the collected rate of 52% with post-secondary education hard to interpret. According to national statistics from Statistics Sweden (*Statistics Sweden, 2010*), this figure is 35.5%. This indicates that our data on educational level may not be valid. A final important limitation of this study is the response rate of 44.3%, as for example compared to *Kessler et al. (2005)* who had 70.9% response rate. Among community surveys using the PHQ-9, our response rate is low in comparison to 64.5% (*Martin et al., 2006*), but higher than in *Pirkis et al. (2009)* who reported 28.6%. This could have biased the prevalence estimates in this study. Importantly, there are indications that individuals with mental illness might be more reluctant than others to participate in mental health surveys (*Allgulander, 1989*; *Eaton et al., 1992*). Based on this, prevalence estimates of depression and anxiety disorders in this study would be underestimated, if biased. Future research with a similar design to this study should make use of established strategies to increase the response rate, for example, by using monetary incentives, using pre-notification and by having a shorter questionnaire (*Edwards et al., 2007*).

Some implications of this study are discussed as follows. The results from this study complements and corroborates earlier findings from Sweden and other countries by providing up-to-date prevalence rates. The fact that 17.2% of the population suffer from a clinically significant depression or anxiety condition calls for further efforts regarding prevention and treatment. This includes both novel and more established intervention alternatives. The use of the Internet is one way of enabling evidence-based interventions to reach larger proportions of the general population (*Muñoz, 2010*). Internet-based psychological treatments seem to be a promising and feasible treatment alternative that may increase availability of evidence-based treatments for psychiatric disorders such as depres-

sion and anxiety (*Andersson, 2009*; *Hedman, Ljótsson & Lindefors, 2012*). Future challenges include establishing means of delivering such treatment in regular health-care (*Hedman et al., 2013*), but also on national level with preserved efficacy (*Calear et al., 2009*).

A further implication regards the significant amount of comorbidity between depression and anxiety disorders identified in this study (see Fig. 1). The large overlap calls for alternatives to disorder-focused assessments and treatments that address transdiagnostic factors, such as emotion regulation deficits (*Aldao, Nolen-Hoeksema & Schweizer, 2010*; *Mennin, McLaughlin & Flanagan, 2009*). This could potentially enable assessment that is grounded in experimental research and that provide an opportunity for treatment alternatives that target these processes (e.g., *Farchione et al., 2012*; *Johansson et al., 2012*; *McEvoy, Nathan & Norton, 2009*). Hence, transdiagnostic treatment approaches seem as an important step in addressing psychiatric comorbidity.

Finally, there are indications that positive mental health likely is more than absence of psychiatric symptoms (*Huppert & Whittington, 2003*). For example, the absence of positive emotions have been found to be a better predictor of mortality than the presence of psychological symptoms (*Huppert & Whittington, 2003*). Also, there are indications that disability and lack of social roles are important determinants of psychological symptoms, but have less influence on positive well-being. Similarly, paid employment has been found to be an important determinant of positive well-being but it seem to have little influence on psychiatric symptoms (*Huppert & Whittington, 2003*). Therefore, future epidemiological research that aims to understand the whole picture of human health seems to require assessments that go beyond mere symptoms (*Cloninger, 2006*; *Huppert & Whittington, 2003*).

## CONCLUSIONS

The findings reported here provide updated prevalence rates from the Swedish general population. Depression, anxiety and their comorbidity are undertreated conditions that are associated with lower quality of life. This study adds to existing evidence that these conditions are clearly major health problems in Sweden, that need further efforts regarding preventive and treatment interventions.

## ACKNOWLEDGEMENTS

We would like to thank Elinore Hellqvist for help during the data collection.

### Funding

This study was sponsored in part by a grant to Gerhard Andersson from Linköping University. The funders had no role in study design, data collection and analysis, decision to publish, or preparation of the manuscript.

### Grant Disclosures

The following grant information was disclosed by the authors:
Linköping University.

## Competing Interests

Gerhard Andersson is an Academic Editor for PeerJ.

## Author Contributions

- Robert Johansson conceived and designed the experiments, analyzed the data, wrote the paper.
- Per Carlbring, Åsa Heedman and Gerhard Andersson conceived and designed the experiments, performed the experiments, analyzed the data, wrote the paper.
- Björn Paxling analyzed the data, wrote the paper.

## Human Ethics

The following information was supplied relating to ethical approvals (i.e. approving body and any reference numbers):

Approval for the study was obtained from the institutional review board at the Department of Behavioural Sciences and Learning, Psychology section, Linköping University, Sweden. The verbal approval was attained during a meeting with the board at Linköping University.

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
