# Peer review of "Depression, anxiety and their comorbidity in the Swedish general population: point prevalence and the effect on health-related quality of life"

_PeerJ, doi:10.7717/peerj.98_

## Round 0.1 · original submission · Minor Revisions

Both reviewers recommended minor revisions. You may use their responses as an indication of how you can make your paper more interesting and useful for readers. Hopefully this will improve your submission to justify acceptance without further revision. I appreciate a clear summary of what changes you have made or not with a justification.

Reviewer 1 ·

Basic reporting

Adequate

Experimental design

Adequate methodology used as well as self report valid instruments.

Validity of the findings

Result and discussion sections can be improved.
Tables can be improved as the number of participants is repeated in the area where only the statistical information whould be included. In all tables discard the "total column" as the main comparisons are according to gender.
Author should include more useful information in the discussion section as many of the results are repeated. Also, very few information about the implications of the results is included. Authors should include information of future lines of research in the area of comorbidity in Swedish population as well as strategies to increase the response rate in this population

Additional comments

Some minor observations were done for manuscript improvement, specially in the result and discussion sections. :

·

Basic reporting

First of all I would like to thank you for the opportunity to review this article. The study presented here is a epidemiological study investigating the prevalence of depression, generalized anxiety disorder and anxiety disorders in general in the Swedish population. The researchers have used adequate self-report measures, instead of clinical interviews, and provide point prevalence rates of depression, anxiety disorders and their comorbidity. The researchers found that these conditions might be undertreated in Sweden, as is in other countries, and are associated with lower quality of life.

At a general level I found the study well executed and the paper organized and informative. I also agree with the authors regarding the necessity of their study, which provides new relevant information that complements and corroborates earlier findings. The authors are also aware of the limitations of their study, thus I will not be pointing these in this review. I have only minor concerns I want to detail next.

INTRODUCTION
I found the introduction informative enough. However, I would appreciate if the authors could also describe more explicit the link between quality of life and depression and anxiety before presenting the reader with the aims of their study.

MATERIALS AND METHODS
I appreciated that the authors described the procedure and sample in great detail. If possible, I suggest the description of the cut-off points for the depression measure to be more straightforward.

RESULTS
I have no comments here. Nice figures!

DISCUSSION
I would like the authors to address the question of other measures that could be needed to get the whole picture of human health. For example, there are indications that the absence of positive emotions are a better predictor of morbidity than the prescence of negative emotions (see Cloninger, 2006, Huppert & Whittington, 2003)

References
Cloninger, C. R. (2006). The science of well-being: an integrated approach to mental health and its disorders. World Psychiatry, 5, 71-76.

Huppert F. A., & Whittington, J. E. (2003). Evidence for the independence of positive and negative well- being: implications for quality of life assessment. British Journal of Health Psychology, 8, 107- 122.

Experimental design

No comments

Validity of the findings

No comments

---

## Round 0.2 · Minor Revisions

It appears you have only responded to the first review and not to the second (and the recently submitted third). Please consider all the reviews in your final revision for my decision.

---

## Round 0.3 · accepted · Accept

Thank you for these revisions, which I think make the limitations and implications of your epidemiological findings more clear. This is a valuable addition to the literature.